# FACT-DRIVEN LOGIC REASONING

## ABSTRACT

Recent years have witnessed an increasing interest in training machines with reasoning ability, which deeply relies on accurate, clearly presented clue forms that are usually modeled as entity-like knowledge in existing studies. However, in real hierarchical reasoning motivated machine reading comprehension, such one-sided modeling is insufficient for those indispensable local complete facts or events when only "global" knowledge is really paid attention to. Thus, in view of language being a complete knowledge/clue carrier, we propose a general formalism to support representing logic units by extracting backbone constituents of the sentence such as the subject-verb-object formed "facts", covering both global and local knowledge pieces that are necessary as the basis for logic reasoning. Beyond building the ad-hoc graphs, we propose a more general and convenient fact-driven approach to construct a supergraph on top of our newly defined fact units, benefiting from both sides of the connections between facts and internal knowledge such as concepts or actions inside a fact. Experiments on two challenging logic reasoning benchmarks show that our proposed model, FOCAL REASONER, outperforms the baseline models dramatically and achieves state-of-the-art results.

## 1 INTRODUCTION

To understand human language, deep neural networks have been widely applied and achieved impressive benchmark results (Chen et al., 2016b; Sachan & Xing, 2016; Seo et al., 2017; Dhingra et al., 2017; Cui et al., 2017; Song et al., 2018; Hu et al., 2019; Zhang et al., 2020a; Back et al., 2020; Zhang et al., 2020b; Hermann et al., 2015). However, the core requirement among the natural language understanding, logic reasoning, cannot be simply solved by the current design philosophy of extracting statistical patterns from data (Shi et al., 2020). In order to solve such a problem, there emerges an interest that accounts for human intuition about the entailment of sentences and reflects the semantic relations between sentential constituents (Iwańska, 1993). In this paper, we focus on logic reasoning in the form of natural language understanding (NLU) as logic reasoning may be naturally embodied in such a task and natural language offers sufficient enough clues for effective logic reasoning. In detail, we concern about a logic reasoning question-answering (QA) task, where given passage, question, and candidate answer options, the model has to make a proper decision with its logic reasoning ability. There are examples shown in Figure 1 from logic reasoning benchmark datasets, ReClor (Yu et al., 2020) and LogiQA (Liu et al., 2020).

Recent neural models usually exploit a pre-trained language model (PrLM) as a key encoder for effective contextualized representation. According to diagnostic tests (Ettinger, 2020; Rogers et al., 2020), though PrLMs like BERT (Devlin et al., 2019) have encoded syntactic and semantic information after large-scale pre-training, they perform sensitivity to role reversal and struggles with pragmatic inference and role-based event knowledge, which are fundamental for reasoning whose major challenge is to uncover logical structures, and reason with the candidate options and questions to predict the correct answer. It is difficult for PrLMs to

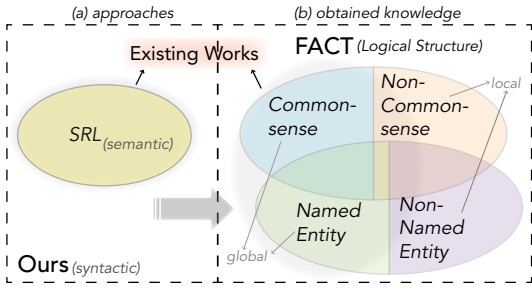

Figure 2: Our "fact" V.S. existing approaches.

| Question | Passage | Answer |
|---|---|---|
| **Example 1**

From this we know | Xiao Wang is taller than Xiao Li, Xiao Zhao is taller than Xiao Qian, Xiao Li is shorter than Xiao Sun, and Xiao Sun is shorter than Xiao Qian. | ✓ A. Xiao Li is shorter than Xiao Zhao.
B. Xiao Wang is taller than Xiao Zhao.
C. Xiao Sun is shorter than Xiao Wang.
D. Xiao Sun is taller than Xiao Zhao. |
| **Example 2**

Which one of the following statements, most seriously weakens the argument? | .... A large enough comet colliding with Earth could have caused a cloud of dust that enshrouded the planet and cooled the climate long enough to result in the dinosaurs' demise. | A. Many other animal species from same era did not become extinct at the same time the dinosaurs did.
B. It cannot be determined from dinosaur skeletons whether the animals died from the effects of a dust cloud.
C. The consequences for vegetation and animals of a comet colliding with Earth are not fully understood.
✓ D. Various species of animals from the same era and similar to them in habitat and physiology did not become extinct. |

Figure 1: Two examples from LogiQA and ReClor respectively are illustrated. There are arguments and relations between arguments. Both are emphasized by different colors: arguments, relations. Key words in questions are highlighted in Purple. Key options are highlighted in gray.

capture the logical structure inherent in the texts since logical supervision is rarely available during pre-training. Existing logic reasoning has shown serious dependence on knowledge-like clues. This is due to the lengthy, noisy text in human language which is though a natural carrier of knowledge but does not provide a clean, exact knowledge form. Thus, an increasing interest is in using graph networks to model the entity-aware relationships in the passages (Yasunaga et al., 2021; Ren & Leskovec, 2020; Huang et al., 2021; Krishna et al., 2020; Lv et al., 2020). However, all these methods may insufficiently capture indispensable logical units from two perspectives. First, they mostly focus on entity-aware commonsense knowledge, but pay little attention to those non-entity, non-commonsense clues (Zhong et al., 2021). Second, when existing models extract predicate logic inside language into knowledge, they only exploit quite limited predicates like *hasA* and *isA* but ignore a broad range of predicates in real language. From either of the perspectives, the existing methods actually only concern about those "global" knowledge that keeps valid across the entire data, without sufficient "local" perception of complete facts or events in the given specific part of logic reasoning task. We argue such insufficient modeling on logic units roots from the ignorance of language itself being the complete knowledge/clue carrier. Thus, we propose extracting a kind of broad *facts* according to backbone constituents of a sentence to effectively cover such indispensable logic reasoning basis, filling the gap of local, non-commonsense, non-entity, or even non-knowledge clues in existing methods as shown in Figure 2. For example, these units may reflect the facts of *who did what to whom*, or *who is what* in Figure 3. Such groups can be defined as "fact unit" following Nakashole & Mitchell (2014) in Definition 1. The fact units are further organized into a supergraph following Definition 2.

**Definition 1** *(Fact Unit) Given an triplet $T = \{E_1, P, E_2\}$, where $E_1$ and $E_2$ are arguments (including entity and non-entity), $P$ is the predicate, a fact unit $F$ is the set of all entities in $T$ and their corresponding relations.*

**Definition 2** *(Supergraph) A supergraph is a structure made of fact units (regarded as subgraphs) as the vertices, and the relations between fact units as undirected edges.*

As shown in Figure 2, we regard the defined *fact* as the results of syntactic processing, rather than those from semantic role labeling (SRL) as in previous study, thus the proposed *fact* also extends the processing means in existing work. Correspondingly, in this work, we propose a fact-driven logical reasoning model, called FOCAL REASONER, which builds supergraphs on top of fact units as the basis for logic reasoning, to capture both global connections between facts and the local concepts or actions inside the fact. Our model FOCAL REASONER is evaluated on two challenging logic reasoning benchmarks including ReClor, LogiQA, one dialogue reasoning dataset Mutual, for verifying the effectiveness and the generalizability across different domains and question formats.

## 2 RELATED WORK

**Machine Reading Comprehension**    Recent years have witnessed massive researches on Machine Reading Comprehension (MRC) whose goal is training machines to understand human languages,

which has become one of the most important areas of NLP (Chen et al., 2016b; Sachan & Xing, 2016; Seo et al., 2017; Dhingra et al., 2017; Cui et al., 2017; Song et al., 2018; Hu et al., 2019; Zhang et al., 2020a; Back et al., 2020; Zhang et al., 2020b). Despite the success of MRC models on various datasets such as CNN/Daily Mail (Hermann et al., 2015), SQuAD (Rajpurkar et al., 2016), RACE (Lai et al., 2017) and so on, researchers began to rethink what extent does the problem been solved. Nowadays, there are massive researches into the reasoning ability of machines. According to (Kaushik & Lipton, 2018; Zhou et al., 2020; Chen et al., 2016a), reasoning abilities can be broadly categorized into (1) commonsense reasoning (Davis & Marcus, 2015; Bhagavatula et al., 2019; Talmor et al., 2019; Huang et al., 2019); (2) numerical reasoning (Dua et al., 2019); (3) multi-hop reasoning (Yang et al., 2018) and (4) logic reasoning (Yu et al., 2020; Liu et al., 2020), among which logic reasoning is essential in human intelligence but has merely been delved into. Natural Language Inference (NLI) (Bowman et al., 2015; Williams et al., 2018; Nie et al., 2020) is a task closely related to logic reasoning. However, it has two obvious drawbacks in measuring logic reasoning abilities. One is that it only has three logical types which are *entailment, contradiction* and *neutral*. The other is its limitation on sentence-level reasoning. Hence, it is important to research more comprehensive and deeper logic reasoning abilities.

**logic reasoning in MRC** There are two main kinds of features in language data that would be the necessary basis for logic reasoning: 1) *knowledge*: global facts that keep consistency regardless of the context, such as commonsense, mostly derived from named entities; 2) *non-knowledge*: local facts or events that may be sensitive to the context, mostly derived from detailed language. Existing works have made progress in improving logic reasoning ability (Yasunaga et al., 2021; Ren & Leskovec, 2020; Huang et al., 2021; Krishna et al., 2020; Zhong et al., 2021; Wang et al., 2021). However, these approaches are barely satisfactory as they mostly focus on the global facts such as typical entity or sentence-level relations, which are obviously not sufficient. In this work, we strengthen the basis for logic reasoning by unifying both types of the features as "facts". Different from previous studies that focus on the knowledge components, we propose a fact-driven logic reasoning framework that builds supergraphs on top of fact units to capture both global connections between entity-aware facts and the local concepts or events inside the fact.

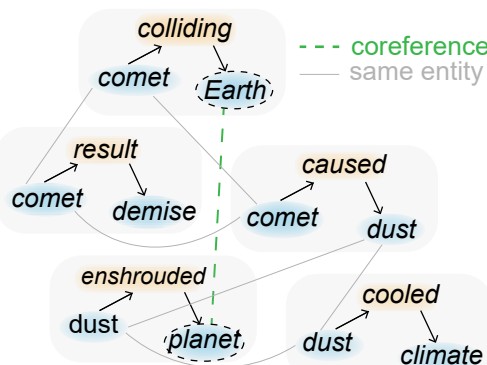

Figure 3: An example of constructed supergraph. In contrast, the dotted vertices and edges are focused in most existing studies (Qiu et al., 2019; Ding et al., 2019; Chen et al., 2019b).

# 3 METHODOLOGY

In this section, we present a fact-driven approach for logic reasoning and the overall architecture of the model is shown in Figure 4. The framework can be divided into three steps as following. We first extract fact units from raw texts via syntactic processing to construct a supergraph. Then it performs reasoning over the supergraph along with a logical fact regularization. Finally, it aggregates the learned representation to decode for the right answer.

## 3.1 FACT UNIT EXTRACTION AND SUPERGRAPH CONSTRUCTION

**Fact Unit Extraction.** Figure 5 illustrates our method for constructing a supergraph from raw text inputs. The first step is to obtain triplets that constitute a fact unit. To keep the framework generic, we use a fairly simple fact unit extractor based on the syntactic relations. Given a context consisting of multiple sentences, we first conduct dependency parsing of each sentence. After that, we extract the subject, the predicate, and the object tokens to get the `"Argument-Predicate-Argument"` triplets corresponding to each sentence in the context.

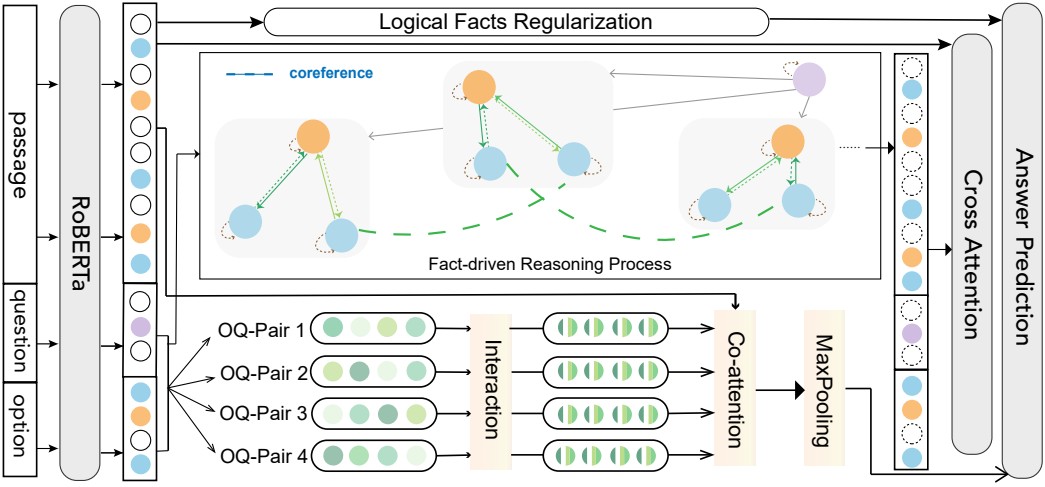

Figure 4: The framework or our model. For supergraph reasoning, in each iteration, each node selectively receives the message from the neighboring nodes to update its representation. The dashed circle means zero vector.

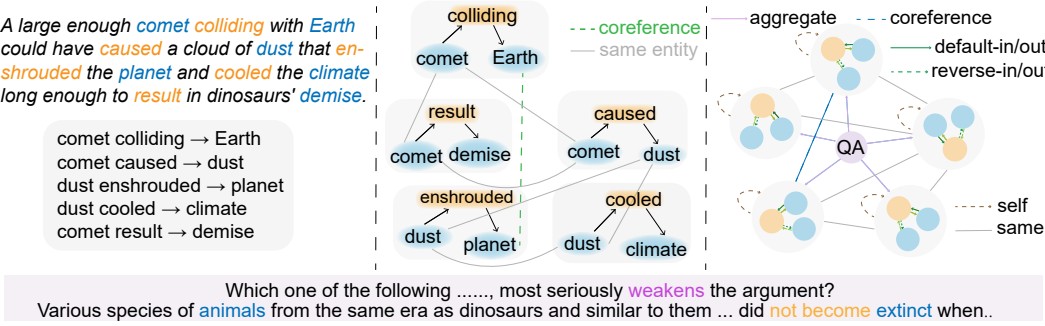

Figure 5: The process of constructing the fact chain and its corresponding Levi graph form of an example in Figure 1. Entities and relations are illustrated in their corresponding color.

**Supergraph Construction.** With the obtained triplets, the fact units are organized in the form of Levi graph (Levi, 1942), which turns arguments and predicates all into nodes. An original fact unit is in the form of $F = (V, E, R)$, where $V$ is the set of the arguments, $E$ is the set of edges connected between arguments, and $R$ is the relations of each edge which are predicates here. The corresponding Levi graph is denoted as $F_l = (V_L, E_L, R_L)$ where $V_L = V \cup R$, which makes the originally directly connected arguments be immediately connected via relations. As for $R_L$, previous works such as (Marcheggiani & Titov, 2017; Beck et al., 2018a) designed three types of edges $R_L = \{default, reverse, self\}$ to enhance information flow. Here in our settings, we extend it into five types: *default-in, default-out, reverse-in, reverse-out, self*, corresponding to the directions of edges towards the predicates. Detailed description for edge types can be found in Appendix A.

We construct the supergraph by making connections between fact units $F_l$. In particular, we take three strategies according to global information, identical concept and co-reference information. (1) We add a node $V_g$ initialized with the question-option representation and connect it to all the fact unit nodes. The edge type is set as *aggregate* for better information interaction. (2) There can be identical mentions in different sentences, resulting in repeated nodes in fact units. We connect nodes corresponding to the same non-pronoun arguments by edges with edge type *same*. (3) We conduct

co-reference resolution on context using an off-to-shelf model[1] in order to identify arguments in fact units that refer to the same one. We add edges with type *coref* between them. The final supergraph is denoted as $S = (F_l \cup V_g, E)$ where $E$ is the set of edges added with the previous three strategies.

## 3.2 REASONING PROCESS

**Graph Reasoning.** A natural way to model the supergraph is via Relational Graph Convolution Networks (Schlichtkrull et al., 2018). We first feed the contexts, stated as $[CLS]C[SEP]q||o[SEP]$, into a pre-trained model to get the encoded representation. We initialize the nodes with averaged hidden states of its tokens because our triplets extraction performs in word-level. For edges, we use a one-hot embedding layer to encode the relations.

Based on the relational graph convolutional network and given the initial representation $h_i^0$ for every node $v_i$, the feed-forward or the message-passing process with information control can be written as $h_i^{(l+1)} = \text{ReLU}(\sum_{r \in R_L} \sum_{v_j \in \mathcal{N}_r(v_i)} g_q^{(l)} \frac{1}{c_{i,r}} w_r^{(l)} h_j^{(l)})$, where $\mathcal{N}_r(v_i)$ denotes the neighbors of node $v_i$ under relation $r$ and $c_{i,r}$ is the number of those nodes. $w_r^{(l)}$ is the learnable parameters of layer $l$. $g_q^{(l)}$ is a gated value between 0 and 1.

Through the graph encoder $F_G(.)$, we then obtain the hidden representations of nodes in fact units as $\{h_0^F, ...h_m^F\} = F_G(\{v_{L,0}, ...v_{L,m}\}, E_L)$. $h_i^F$ is the node representation inside fact unit. They are then concatenated as the representation for supernode as $h_0^S$. $\{h_0, ...h_m\} = F_G(\{h_0^S, ...h_m^S\}, E_C)$.

For node features on the supergraph, it is fused via the attention and gating mechanisms with the original representations of the context encoder $H^C$. We apply attention mechanism to append the supergraph representation to the original one $\tilde{H} = \text{Attn}(H^c, K_f, V_f)$, where $\{K_f, V_f\}$ are packed from the learned representations of the supergraph. We compute $\lambda \in [0, 1]$ to weigh the expected importance of supergraph representation of each source word $\lambda_1 = \sigma(W_\lambda \tilde{H} + U_\lambda H^C)$, where $W_\lambda$ and $U_\lambda$ are learnable parameters. $H^C$ and $\tilde{H}$ are then fused for an effective representation $H = H^C + \lambda \tilde{H} \in \mathbb{R}^{4 \times d}$.

**Interaction.** For the application to the concerned QA tasks that require reasoning, options have their inherent logical relations, which can be leveraged to aid answer prediction. Inspired by Ran et al. (2019), we use an attention-based mechanism to gather option correlation information.

Specifically for an option $O_i$, the information it get by interaction with option $O_j$ is calculated as $O_i^{(j)} = [O_i^q - O_i^q \text{Attn}(O_i^q, O_j^q; v); O_i^q \circ O_i^q \text{Attn}(O_i^q, O_j^q; v)]$, where $O_i^q$ is the representation of the concatenation for the $i$-th option and question after the context encoder. Then the option-wise information are gathered to fuse the option correlation information $\hat{O}_i = \tanh(W_c[O_i^q; \{O_i^{(j)}\}_{i \neq j}] + b_c)$, where $\mathbf{W}_c \in \mathbb{R}^{d \times 7d}$ and $b_c \in \mathbb{R}^d$.

For answer prediction, We seek to minimize the cross entropy loss by $\mathcal{L}_{ans} = -\log softmax(W_z C + b_z)_l \in \mathbb{R}^4$, where $C$ is the combined representations of $\hat{O}$ and $H$.

**Logical Fact Regularization.** Since the subject, verb, and object in a fact should be closely related with some explicit relationships, we design logical fact regularization technique to make the logical facts more of factual correctness. Without loss of generality, we assume that in our settings, the summation of the subject vector and the relation vector should be close to the object vector as much as possible, i.e., $v_{subject} + v_{predicate} \to v_{object}$. Specifically, given the hidden states of the sequence $h_i$ where $i = 1, \ldots, L$ and $L$ is the total length of the sequence, The regularization is defined as $L_{lfr} = \sum_{k=1}^m (1 - \cos(h_{sub_k} + h_{pred_k}, h_{obj_k}))$, where $m$ is the total number of logical fact triplets extracted from the context as well as the option and $k$ indicates the $k$-th fact triplet.

## 3.3 TRAINING OBJECTIVE

During training, the overall loss for answer prediction is $\mathcal{L} = \alpha \mathcal{L}_{ans} + \beta \mathcal{L}_{lfr}$, where $\alpha$ and $\beta$ are two parameters. In our implementation, we set $\alpha = 1.0$ and $\beta = 0.5$.

---

[1] https://github.com/huggingface/neuralcoref.

Table 1: Experimental results of our model compared with baseline models on ReClor and LogiQA dataset. Test-E and Test-H denote Test-Easy and Test-Hard respectively. The results in **bold** are the best performance except for the human performance. * indicates that the results are taken from Yu et al. (2020) and Liu et al. (2020). Results with † is taken from Wang et al. (2021).

| Model | ReClor | | | | LogiQA | |
|---|---|---|---|---|---|---|
| | Dev | Test | Test-E | Test-H | Dev | Test |
| Human Performance* | - | 63.0 | 57.1 | 67.2 | - | 86.0 |
| BERT* | 53.8 | 49.8 | 72.0 | 32.3 | 34.1 | 31.0 |
| XLNet* | 62.0 | 56.0 | 75.7 | 40.5 | - | - |
| RoBERTa* | 62.6 | 55.6 | 75.5 | 40.0 | 35.0 | 35.3 |
| DeBERTa† | 74.4 | 68.9 | 83.4 | 57.5 | 44.4 | 41.5 |
| DAGN$_{\mathrm{RoBERTa}}$* | 65.8 | 58.3 | 75.9 | 44.5 | 36.9 | 39.3 |
| - data augmentation | 65.2 | 58.2 | 76.1 | 44.1 | 35.5 | 38.7 |
| LReasoner$_{\mathrm{RoBERTa}}$† | 66.2 | 62.4 | 81.4 | 47.5 | 38.1 | 40.6 |
| - data augmentation | 65.2 | 58.3 | 78.6 | 42.3 | - | - |
| LReasoner$_{\mathrm{DeBERTa}}$† | 74.6 | 71.8 | 83.4 | 62.7 | 45.8 | 43.3 |
| FOCAL REASONER$_{\mathrm{RoBERTa}}$ | 66.8(↑4.2) | 58.9(↑3.3) | 77.1(↑1.6) | 44.6(↑4.6) | 41.0(↑6.0) | 40.3(↑5.0) |
| FOCAL REASONER$_{\mathrm{DeBERTa}}$ | **78.6**(↑4.2) | **73.3**(↑4.4) | **86.4**(↑3.0) | **63.0**(↑5.5) | **47.3**(↑2.9) | **45.8**(↑4.3) |

## 4 EXPERIMENTS

### 4.1 EXPERIMENTAL SETUP

We conducted the experiments on three datasets. Two for specialized logic reasoning ability testing: ReClor (Yu et al., 2020) and LogiQA (Liu et al., 2020) and one for logic reasoning in dialogues: MuTual (Cui et al., 2020).

We take RoBERTa-large (Liu et al., 2019) and DeBERTa-xlarge (He et al., 2020) as our backbone models for convenient comparison with previous works. We also compare our model with DAGN (Huang et al., 2021), a framework leveraging RoBERTa-large as the backbone and LReasoner (Wang et al., 2021), the previous state-of-the-art model on the leaderboard using DeBERTa-xlarge. For more details on datasets and baseline models, one can refer to Appendix A.

### 4.2 IMPLEMENTATION DETAILS

The overall model is end-to-end trained and updated by Adam (Kingma & Ba, 2015) optimizer with an overall learning rate 8e-6 for ReClor and LogiQA, and 4e-6 for MuTual. The weight decay is 0.01. We set the warm-up proportion during training to 0.1. Graph encoders are implemented using DGL, an open-source lib of python. The layer number of the graph encoder is 2 for ReClor and 3 for LogiQA. The maximum sequence length is 256 for LogiQA and MuTual, and 384 for ReClor. The model is trained for 10 epochs with a total batch size 16 and an overall dropout rate 0.1 on 4 NVIDIA Tesla V100 GPUs, which takes around 2 hours for ReClor and 4 hours for LogiQA[2].

### 4.3 RESULTS

Tables 1 and 2 show the results on ReClor, LogiQA, and MuTual, respectively. All the best results are shown in bold. From the results, we have the following observations:

1) Based on our implemented baseline models (basically consistent with public results), we observe dramatic improvements on both of the logic reasoning benchmarks, e.g., on ReClor test set, FOCAL REASONER achieves +4.2% on dev set and +3.3% on the test set. FOCAL REASONER also outperforms the prior best system LReasoner[3], reaching 77.05% on the EASY subset, and 44.64% on the HARD subset. The performance suggests that FOCAL REASONER makes better use of logical

---

[2]Our code has been submitted along with this submission, which will be open after the blind review period.

[3]The test results are from the official leaderboard `https://eval.ai/web/challenges/challenge-page/503/leaderboard/1347`.

Table 2: Experimental results of our model compared with baseline on MuTual dataset. * indicates that the results are taken from Cui et al. (2020). For fair comparison with our method, we also report the multi-choice method (RoBERTa-MC) in addition to the default Individual scoring method (RoBERTa).

| Model | MuTual | | | | | |
| | Dev Set | | | Test Set | | |
| | $R_4@1$ | $R_4@2$ | MRR | $R_4@1$ | $R_4@2$ | MRR |
|---|---|---|---|---|---|---|
| RoBERTa* | 69.5 | 87.8 | 82.4 | 71.3 | 89.2 | 83.6 |
| RoBERTa-MC* | 69.3 | 88.7 | 82.5 | 68.6 | 88.7 | 82.2 |
| FOCAL REASONER | **73.4**(↑4.1) | **90.3**(↑1.6) | **84.9**(↑2.4) | **72.7**(↑4.1) | **91.0**(↑2.3) | **84.6**(↑2.4) |

Table 3: Accuracy on the dev set of ReClor corresponding to several representative question types. *S: Strengthen, W: Weaken, I: Implication, CMP: Conclusion/Main Point, ER: Explain or Resolve, D: Dispute, R: Role, IF: Identify a Flaw, MS: Match Structures*. All results are reported on the same PrLM RoBERTa.

| Model | S | W | I | CMP | ER | P | D | R | IF | MS |
|---|---|---|---|---|---|---|---|---|---|---|
| RoBERTa | 61.7 | 47.8 | 39.1 | 63.9 | 58.3 | 50.8 | 50.0 | 56.3 | 61.5 | 56.7 |
| DAGN | 63.8 | 46.0 | 39.1 | 69.4 | 57.1 | 53.9 | 46.7 | 62.5 | 62.4 | 56.7 |
| FOCAL REASONER | 72.3 | 66.4 | 47.8 | 91.7 | 76.2 | 76.9 | 66.7 | 68.8 | 73.5 | 86.7 |

structure inherent in the given context to perform reasoning than existing methods. More detailed ablations will be shown in Section 5.

2) Table 3 lists the accuracy of our model on the dev set of ReClor of different question types. Results show that our model can perform well on most of the question types, especially "Strengthen" and "Weaken". This means that our model can well interpret the question type from the question statement and make the correct choice corresponding to the question.

3) Our model also achieves comparable performance with the unpublished LReasoner. It employs symbolic reasoning and data augmentation techniques, which is in a different research line from ours. Without data argumentation, LReasoner shows relatively poorer results, showing that our fact-driven approach would be beneficial compared with the symbolic-driven technique. Compared with the neural methods for logic reasoning, symbolic approaches would rely heavily on dataset-related predefined patterns which entail massive manual labor, potentially restricting the generalizability of models.

4) On the dialogue reasoning dataset MuTual, our model achieves quite a jump compared with the RoBERTa-base LM.[4] This verifies our model's generalizability on other downstream reasoning task settings.

5) For the model complexity, our method basically keeps as simple as previous models like DAGN. Our model only has 414M parameters compared with 355M in the baseline RoBERTa, and 400M in DAGN which also employs GNN.

## 5 ANALYSIS

### 5.1 ABLATION STUDY

To dive into the effectiveness of different components in FOCAL REASONER, we conduct analysis by taking RoBERTa as the backbone on the ReClor dev set. Tables 5-6 summarize the results.

---

[4]Since there are no official results on RoBERTa-large LM, we use RoBERTa-base LM instead for consistency.

Table 4: Statistics for fact unit entities and traditional named entities in datasets.

| Number | ReClor | | LogiQA | |
|---|---|---|---|---|
| | Train | Dev | Train | Dev |
| Fact Unit Argument | 14,895 | 1,665 | 20,676 | 1,981 |
| Named Entity | 9,495 | 984 | 12,439 | 1,515 |

Table 5: Replacing fact units with named entities or semantic roles on the ReClor dev set.

| Model | Accuracy |
|---|---|
| FOCAL REASONER | $66.8_{\pm 0.13}$ |
| w/ named entity only | $62.8_{\pm 0.26}$ |
| w/ semantic role only | $62.2_{\pm 0.32}$ |

**Fact Units Variants**   Apart from our syntactically constructed fact units, there are two other ways in different granularities for construction. We replace the fact units with named entities that are used in previous works like Chen et al. (2019a). The statistics of fact units and named entities of ReClor and LogiQA are stated in Table 4, from which we can infer that there are indeed more fact units than named entities. Thus using fact units can better incorporate the logical information within the context. When replacing all the fact units with named entities, we can see from Table 5 that it significantly decreases the performance. We also explore the performance using semantic role labeling a similar way as in Zhong et al. (2020). We can see that SRL, leveraging much more complex information as well as computation complexity, fails to achieve performance as good as our original fact unit.

**Supergraph Reasoning:**   The first key component is supergraph reasoning. We ablate the global atom and erase all the edges connected with it. The results suggest that the global atom indeed betters message propagation, leveraging performance from $64.6\%$ to $66.8\%$. We also find that replacing the initial QA pair representation of the global atom with only question representation hurts the performance. In addition, without the logical fact regularization, the performance drops from $66.8\%$ to $64.2\%$, indicating its usefulness. For edge analysis, when (1) all edges are regarded as a single type rather than the original designed 8 types in total and (2) co-reference edges are removed, the accuracy drops to $63.7\%$ and $64.8\%$, respectively. It is proved that in our supergraph, edges link the fact units in reasonable manners, which properly uncovers the logical structures.

Table 6: Ablation results on the ReClor dev set.

| Model | Accuracy |
|---|---|
| FOCAL REASONER | $66.8_{\pm 0.13}$ |
| **Supergraph Reasoning** | |
| - global edge | $64.6_{\pm 0.32}$ |
| - co-reference edges | $64.8_{\pm 0.24}$ |
| - logical fact regularization | $64.2_{\pm 0.12}$ |
| - edge type | $63.7_{\pm 0.19}$ |
| **Interactions** | |
| - interactions | $65.5_{\pm 0.52}$ |

**Interactions:**   We further experimented with the query-option-interactions setting to see how it affects the performance. The results suggest that the features learned from the interaction process enhance the model. Considering that the logical relations between different options are a strong indicator of the right answer, this means that the model learns from a comparative reasoning strategy.

## 5.2   EFFECTS OF FACT UNITS NUMBERS

To inspect the effects of the number of fact units, we split the original dev set of ReClor and LogiQA into 5 subsets. The statistics of the fact unit distribution on the datasets are shown in Table 7. The numbers of fact units for most contexts in ReClor and LogiQA are in $[3, 6)$ and $[0, 3)$, respectively.

Comparing the accuracies of RoBERTa-large baseline, prior SOTA LReasoner and our proposed FOCAL REASONER in Figure 6, our model outperforms baseline models on all the divided subsets, which demonstrates the effectiveness and robustness of our proposed method. Specifically, for ReClor, FOCAL REASONER performers better when there are more fact units in the context, while for LogiQA, FOCAL REASONER works better when the number of fact units locates in $[0, 3)$ and $[9, 12)$. The reason may lie in the difference in style of the two datasets. However, all the models include ours struggle when the number of fact units is above certain thresholds, i.e., the logical structure is more complicated, calling for better mechanisms to cope with.

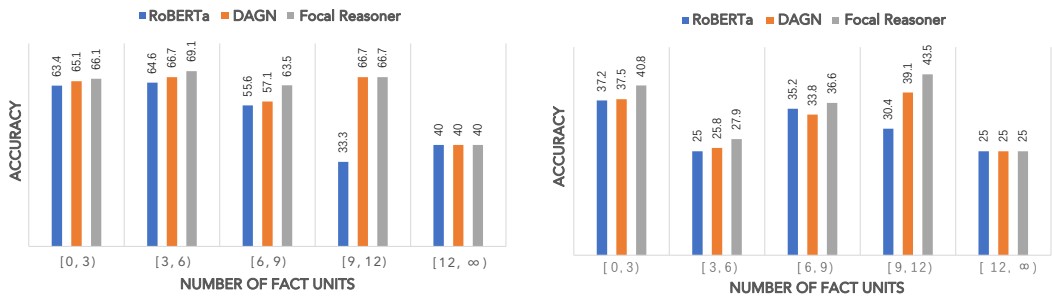

Figure 6: Accuracy of models on number of fact units on dev set of ReClor (left) and LogiQA (right).

## 5.3 INTERPRETABILITY: A CASE STUDY

We aim to interpret FOCAL REASONER's reasoning process by analyzing the node-to-node attention weights induced in the supergraph in Figure 7. We can see that our FOCAL REASONER can well bridge the reasoning process between context, question and option. Specifically, in the graph, "students rank 30%" attends strongly to "playing improve performance". Under the guidance of question to select the option that weakens the statement and option interaction, our model is able to tell that "students rank 30% can play" mostly undermines the conclusion that "playing improves performance".

Table 7: Distribution of fact unit number on dev set of the training datasets.

| Dataset | $[0, 3)$ | $[3, 6)$ | $[6, 9)$ | $[9, 12)$ | $[12, \infty)$ |
|---------|----------|----------|----------|-----------|----------------|
| ReClor  | 37.2%    | 48.6%    | 12.6%    | 0.6%      | 1.2%           |
| LogiQA  | 47.5%    | 37.5%    | 10.9%    | 3.5%      | 0.6%           |

A recent survey in a key middle school showed that high school students in this school have a special preference for playing football, and it far surpasses other balls.The survey also found that students who regularly play football are better at academic performance than students who do not often play football.This shows that often playing football can improve students' academic performance.

✓ A. Only high school students who are ranked in the top 30% of grades can often play football.
B. Regular football can exercise and maintain a strong learning energy.
C. Often playing football delays the study time.
D. Research has not proved that playing football can contribute to intellectual development.

Which of the following can weaken the above conclusion most?

① 1. students have preferences
2. preference playing football
3. it surpasses balls
4. who play football
5. students better performance
6. who !play football
7. playing improve performance
8. students rank 30%
9. students play football

**Fact Units**

② QA

Edge Weight
0     1

③ Option Similarity Matrix after Interaction

✓A: 4.1918    C: -12.3718
B: -5.3050    D: -6.9722

Figure 7: An example of how our model reasons to get the final answer.

## 6 CONCLUSION

For logic reasoning arising from machine reading comprehension, it is well known that clear and accurate forms like global knowledge are crucial. In this work, we make a finding that existing studies miss focusing on quite a lot of non-knowledge parts which is also indispensable for better reasoning. Thus we propose extracting a general form called "fact unit" to cover both global and local logical units, hoping to shed light on the basis of structural modeling for logic reasoning. Our proposed FOCAL REASONER not only better uncovers the logical structures within the context, which can be a general method for other sophisticated reasoning tasks, but also better captures the logical interactions between context and options. The experimental results verify the effectiveness of our method.

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

## A    LEVI GRAPH CONSTRUCTION

Levi graph transformation turns labeled edges into additional vertices. There are two types of edges in a traditional Levi graph: *default* and *reverse*. For example, an edge $(E1, R1, E2)$ in the original graph becomes $(E1, default, R1)$, $(R1, default, E2)$, $(R1, reverse, E1)$ and $(E2, reverse, R1)$.

However, the type of source and target vertices in the graph also matters (Beck et al., 2018b). Specifically, previous works use the same type of edge to pass information, which may reduce the effectiveness. Thus we propose to transform the *default* edges into *default-in* and *default-out* edges, and the *reverse* edges into *reverse-in* and *reverse-out* edges.

## B    DETAILS FOR DATASETS AND BASELINE MODELS

In this section, we describe the datasets and baseline models used in the experiments.

### B.1    DATASETS

**ReClor**    ReClor contains 6,138 multiple-choice questions modified from standardized tests such as GMAT and LSAT, which are randomly split into train/dev/test sets with 4,638/500/1,000 samples respectively. It contains multiple logic reasoning types. The held-out test set is further divided into EASY and HARD subsets based on the performance of BERT-based model Devlin et al. (2019).

**LogiQA**    LogiQA consists of 8,678 multiple-choice questions collected from National Civil Servants Examinations of China and are manually translated into English by experts. The dataset is randomly split into train/dev/test sets with 7,376/651/651 samples correspondingly. LogiQA also contains various logic reasoning types.

**MuTual**    MuTual has 8,860 dialogues annotated by linguist experts and high-quality annotators from Chinese high school English listening comprehension test data. It is randomly split into train/dev/test sets with 7,088/886/886 samples respectively. There more than 6 types of reasoning abilities reflected in MuTual. MuTual$^{plus}$ is an advanced version, where one of the candidate responses is replaced by a safe response (e.g., *"could you repeat that?"*) for each example.

### B.2    BASELINE MODELS

**DAGN**    explores passage-level discourse-aware clues used for solving logical reasoning QA. Specifically, they leverage discourse relations annotated in Penn Discourse TreeBank 2.0 (PDTB 2.0) (Prasad et al., 2008) and punctuation as the delimiters to split the context into elementary discourse units (EDUs). They are further organized into a logical graph and feed into a GNN to get the representation.

**LReasoner**    is a symbolic-driven framework for logical reasoning of text. It firstly identifies the logical symbols and expressions explicitly for the context and options based on manually designed rules. Then it performs logical inference over the expressions according to logical equivalence laws such as *contraposition* (Russell & Norvig, 2002). Finally, it verbalizes the expressions to match the answer.

FOCAL REASONER enjoys two major merits. (1) **Broader knowledge:** Compared with DAGN which uses sentential knowledge such as logical connectives (e.g., *becaus*, *however*), FOCAL REASONER leverages a broader type of knowledge characterized by "fact unit", including global knowledge and local knowledge. (2) **More transferable:** Compared with LReasoner which manually design rules to extract logical patterns and perform logical reasoning in a symbolic way, Focal Reasoner is neural-based and manual-free, which is more generalizable to other datasets.

## C    MODEL COMPLEXITY

In this section, we display the statistics for the parameters of FOCAL REASONER and baseline models to demonstrate the model complexity and the effectiveness.

From Table 8 we can see that the increase of parameters is no more than 15% compared with the baseline models, and is comparable with strong baseline model DAGN. This indicates that, our model, being quite effective, is not a result of stacking complicated modules.

Table 8: Distribution of fact unit number on dev set of the training datasets.

| RoBERTa | DAGN | Focal Reasoner |
|---------|------|----------------|
| 355M    | 395M | 409M           |

## D    Variances for Focal Reasoner w.r.t experiment results

In this section, we report the average and variances run on 5 random seeds for FOCAL REASONER with different pre-trained language models.

Table 9: Experimental results for FOCAL REASONER with average results and variances run no 5 random seeds.

| Model | ReClor | | | | LogiQA | |
|-------|--------|------|--------|--------|--------|------|
|       | Dev    | Test | Test-E | Test-H | Dev    | Test |
| FOCAL REASONER$_{\text{RoBERTa}}$ | 66.8±0.13 | 58.8±0.14 | 76.9±0.16 | 44.5±0.12 | 41.0±0.11 | 40.3±0.15 |
| FOCAL REASONER$_{\text{DeBERTa}}$ | 78.6±0.18 | 73.2±0.17 | 86.2±0.21 | 62.9±0.13 | 47.3±0.16 | 45.8±0.17 |

## E    Further Interpretation

We change the example in Figure 7 a bit. Specifically, we change the conclusion from "playing football can improve students' academic performance" to "football players are held to higher academic standards than non-athletes". We can observe that our model can select the relatively correct answer, which indicates that FOCAL REASONER has some logical reasoning ability, instead of simple text matching and mining ability.

Figure 8: An example of how our model reasons to get the final answer on the modified example.

