# OpenReview forum: "Fact-driven Logical Reasoning"
_ICLR.cc/2022/Conference — ICLR 2022 Submitted_

### Official Review · Reviewer_MQGH · 2021-10-31

**Correctness:** 3
**Technical Novelty And Significance:** 3
**Empirical Novelty And Significance:** 2
**Recommendation:** 6
**Confidence:** 3

**Main Review:**

Strengths:
The paper is well-written with clear motivations and structure. The concept of fact units is interesting and novel which are easily constructed via dependency trees. Compared with commonsense and entity-relation knowledge, the fact units are more informative to each specific question. By associating nodes across different fact units based on coreferences and mentions, a supergraph is built that connects all related information and conducts graph reasoning for answer predictions. The proposed model is then experimentally verified on three logic-driven datasets which demonstrates some performance gain.

Weaknesses:
1. Some of the details for model description are missing and confusing, e.g., (1) How the representation is initialized for a supernode corresponding to a fact triplet and how does the supernode propagate information to the sub-nodes? (2) Equation 2 and 3 use the same graph encoder $F_{G}$. Are they the same? Since they both use $m$ nodes, does it mean the supergraph also operates on each sub fact node? If so, does the global node $V_g$ connect to all the sub fact nodes? It is then not clear how each fact block (grey squares in Figure 4) functions as a whole. The notations in equation 2 and 3 are also inconsistent with equation 5. What does L, S, F and m represent?  (3) The loss for answer prediction in Eq 6 is not clearly described: do you use an aggregated embedding over all nodes for prediction? (4) How exactly is the interaction module processed?
2. According to the description, the fact units are constructed using the dependency parser. Hence, the model relies on whether the parser accurately discovers the crucial information. I am also wondering if the extracted facts could bring too much noise to the question. As shown in Figure 6, when more than 12 facts are constructed, the performance becomes worse.
3. More discussions on comparing with symbolic logic reasoner model LReasoner are needed. Indeed, the proposed graph model seems to only implicitly convey knowledge across facts in terms of local reasoning. On the other hand, the symbolic logic rules are able to express global patterns. Hence, to my understanding, it may not be suitable to use the term "global reasoning" in this work.
4. For experiments, Table 1 shows the FOCAL Reasoner (DeBERTa) brings much performance gain, compared to FOCAL Reasoner (RoBERTa) (the performance seems comparable with DAGN and LReasoner using RoBERTa). Can you explain why? Is it possible to run DAGN (DeBERTa)?

**Summary Of The Paper:**

This paper presents a novel graph reasoning model based on supergraphs constructed via fact triplets. The authors define the fact triplets as subject-predicate-object relational path extracted from the dependency parser and use them as local knowledge, compared with the commonly used entity-relation knowledge to enhance the task of logic-driven question answering.

**Summary Of The Review:**

Novel concept of fact units to answer logic-driven questions. More discussions are needed.

---

> ### Author Response · Authors · 2021-11-12
> **Response to Reviewer MQGH**
>
> **W1:** Some of the details of the model are missing and confusing.
>
> **A:** Thanks for your careful review. Sorry for the confusion caused due to the page limit. We have reorganized the Methodology section to provide more details. (1) The representation for a supernode in fact-triplet is initialized by fetching the corresponding token representation from the contextualized sequence representation from the PrLM. (2) $F_G$ denotes that they have the same architecture but are not the same one. (3) $L_{ans}$ is calculated via attention mechanism to fuse the supergraph representation with the original token representation. (4) For interaction module, it compares options to better identify their correlations to help reasoning. In detail, each option is encoded into a vector sequence $O_i$, and an attention mechanism is leveraged [1] to compare these sequences vector-by-vector to identify more subtle correlations with other options $O_j$ to get the interacted representation $O_i^{(j)}$. Then the option-wise information are gathered to fuse the option correlation information $:=tanh(W_c[O_i^q;\{O_i^j\}_{(i\neq j)}]+b_c)$
>
> [1] Qiu Ran, Peng Li, Weiwei Hu, and Jie Zhou. Option comparison network for multiple-choice reading
> comprehension. arXiv preprint arXiv:1903.03033, 2019.
>
> **W2:** Noise brought by fact units.
>
> **A:** Thanks for the insightful observation. The phenomenon of performance getting worse as the number of fact units grows would be that the context becomes more complicated, and the logical relations within become entangled for the model to address. Still, our model surpasses the other counterparts. The accuracy of the dependency parser would not be a major issue since it has been already high enough, and has been widely used in existing studies for annotation [1,2].
>
> [1] Wu, Zhaofeng, Hao Peng, and Noah A. Smith. Infusing finetuning with semantic dependencies. Transactions of the Association for Computational Linguistics 9 (2021): 226-242.
>
> [2] Zhang, Zhuosheng, Yuwei Wu, Junru Zhou, Sufeng Duan, Hai Zhao, and Rui Wang. SG-Net: Syntax-guided machine reading comprehension. In AAAI. Apr 2020.
>
> **W3:** More discussions with LReasoner and the definition of “global reasoning”.
>
> **A:** Thanks for your suggestion. We are willing to add more discussions with LReasoner, however, LReasoner has not been executable yet according to the official GitHub repository, as other researchers discussed in their GitHub issues. We have tried our best to analyze the model with existing statistics from LReasoner to verify the superiority of Focal Reasoner. Of course, we will be very glad to add further analysis with LReasoner once their Github is ready.
> We didn’t use the term “global reasoning” in this work. On the other hand, we use “global knowledge” to highlight that the knowledge captured by fact units such as typical entity or sentence relation.
>
> **W4:** DeBERTa brings much performance gain than other PrLMs.
>
> **A:** The most plausible reason would be that the existing methods on RoBERTa take advantage of other augmentations, such as data augmentation. For DeBERTa, the comparison would be fairer, without further enhancements. We will add the experiments using DeBERTa for DAGN for further analysis once the experiments are finished.

---

> > ### Comment · Reviewer_MQGH · 2021-11-29
> > **Acknowledge of authors' response**
> >
> > I appreciate further explanations from the authors. I will keep my score as it is.

---

### Official Review · Reviewer_UKpM · 2021-11-03

**Correctness:** 4
**Technical Novelty And Significance:** 3
**Empirical Novelty And Significance:** 3
**Recommendation:** 6
**Confidence:** 4

**Main Review:**

As I served as one of the reviewers for this paper before, I am also sharing a summary or quote of reviews from the last round here as the paper did not go through major changes since the last submission. Content-wise, $LReasoner_{RoBERTa}$ , $LReasoner_{DeBERTa}$ , as well as $FocalReasoner_{DeBERTa}$ were added as new result in Table 1.

### Strength

- (From the previous review) All reviewers noted that the main strength of the work to be its good empirical results on the 3 benchmarks considered.
- This strength is corroborated with the additional results of competitive baselines and  $FocalReasoner_{DeBERTa}$.
- The paper is well-written overall. I appreciate the authors making an effort to organize and clarify the complex steps that this paper takes.

### Weakness
-  (From the previous review) The unanimous criticism was that while the results are impressive, given the complex nature of the proposed system, it was difficult to understand which aspects of the proposed techniques are truly effective.
- While the coreference, and entity linking are the major resource in creating the supergraph, I don't think the challenges nor machinery in getting this information is well described.  (It might be in the text, but I am not sure how the authors get dependency parse trees as well.) It would be nice to have  more description on the challenges of identifying the global relations (same entity, coreference)
- As described in the *summary* section, I think the main contribution of this paper is in getting fine-grained facts and connecting them with global information. I am not entirely convinced whether this is a very novel approach. What is the major scientific contribution here?
- The authors attempt to decompose some parts of the method in Tables 5 and 6. Related to the earlier weakness on the complex nature of this work, I am not sure what role the *logical fact regularization loss* serves in this paper. Is this one of the contributions of this paper? Or is it simply a tool to get a better results? Without it, the performance 66.8 --> 64.2 which is lower than competing models.

Some of the comments that other reviewers made from the last round of review (which resonated with me):
- " **Significance**: This is where I am most conflicted about the paper. The results are difficult to interpret. Certainly, there are gains from the paper's technique and the ablations show that the different components of the model all contribute to the performance. But when fairly important elements of the model like coreference edges contribute only around 2% accuracy, it's hard to know how much to trust the narrative here about reasoning."
- "More generally, in a multiple-choice task, it's hard to know that the model is really behaving as advertised rather than just adding capacity on top of the baselines. It's not easy to form an apples-to-apples comparison in terms of number of parameters. The case study in Section 5.3 shows that something is happening in the graph component, but whether this is *causally associated* with the ability to get the right answer as opposed to a byproduct is hard to determine."

### Questions/Suggestions

- I am not sure what each edges really do: (*default-in, default-out, reverse-in, reverse-out, self*). It would be nice if authors can describe the roles of these edges.
- Figure 5's *global edge* and what this paper claims as *global (typical entity or sentence relation)* is confusing. Maybe try to use distinguishable terms?
- Why don't you update 4.3 (1) sentence to match newly updated best result table 1?
    - "FOCAL REASONER also outperforms the prior best system LReasoner, reaching 77:05% on the EASY subset, and 44:64% on the HARD subset."

**Summary Of The Paper:**

The paper claims that a lot of previous work in QA-based MRC focused only on entity-aware common sense knowledge.  To overcome this, the paper proposes to build a finer-grained local fact, (entity, predicate, entity) triplet, information based on dependency parsing, and then connect the triplet nodes with global information such as coreference, entity information.
Architecture-wise, a passage, question, and options are passed to RoBERTa (or DeBERTa) and then this information goes through the graph-attention network (GAN), based on the graph structure described above. In the experiment section, the authors show that the proposed FOCAL REASONER outperforms the previous SOTA models and run an ablation study on how each component contributes (in Table 5,6).

The main contribution of this paper, as I understand, is bringing more fine-grained local facts and connecting them with global information such as coreference and entity linking.

**Summary Of The Review:**

FocalReasoner exhibits a strong experimental result and it is corroborated in the new submission with added experiment results in $LReasoner_{RoBERTa}$ , $LReasoner_{DeBERTa}$ , as well as $FocalReasoner_{DeBERTa}$ in Table 1.

Albeit its high-performance, as the previous review pointed out, I am not sure what is the major factor in the improvement when many parts of the neural machinery are used together. I view the main contribution as using the Levi graph from dependency trees as  Beck et al. 2018 and connecting them with coreference and entity linking.

Overall, I'm borderline about this paper. I think this paper tests whether fine-grained pieces of information in the paragraph are useful or not (intuitively they should, and this paper shows that it does). However, I am not sure whether the usage graphs are novel or significantly better than previous work.

---

> ### Author Response · Authors · 2021-11-12
> **Response to Reviewer UKpM**
>
> **W1:** while the results are impressive, given the complex nature of the proposed system, it was difficult to understand which aspects of the proposed techniques are truly effective. (Significance and complexity)
>
> **A:** Thanks for your review. As we have commented earlier in the last round of review, our goal is not to stack complex modules. Instead, to give full play of “fact unit”, such an architecture has to be designed. We also demonstrate in Section 5 that the design of the “fact unit” is the crucial reason for the performance of our system.
> 1.	We build supergraphs on top of the fact units from syntactic processing, to capture both global connections between facts. To model the supergraph, a natural solution is using GNN.
> 2.	Since the arguments and predicate should be closely related with some explicit relationships to guarantee the quality [1], the logical fact regularization technique is designed to help model the local concepts or actions inside a fact.
>
> Also, our model is not complex as thought from the perspective of model parameters. The number of parameters of our model is around a 15% increase from the original RoBERTa model, which is on par with DAGN. Given the necessary part of the graph neural network, our model architecture is not complicated as thought.
>
> [1] Garcia-Duran, Alberto, Antoine Bordes, and Nicolas Usunier. Composing relationships with translations. Diss. CNRS, Heudiasyc, 2015.
>
> **W2:** It would be nice to have more description on the challenges of identifying the global relations (same entity, coreference)
>
> **A:** Thanks for the suggestion, we will add them in the Appendix.
>
> **W3:** Major scientific contribution.
>
> **A:** The major scientific contribution of this paper can be viewed in two aspects.
> 1. In theory, we innovate the concept of “fact unit”, which is easy-to-obtain but effective enough to act as the knowledge basis for logical reasoning compared with previous knowledge such as SRL and NE.
>
> 2. For the technique, we propose to organize the fact units by supergraph, which is a hierarchical modeling method. Compared to existing methods that only emphasize the relations inside units (SRL/NE) or the relations across units [1], our model has better reasoning ability by modeling the relations inside and across fact units in a systematic way.
>
> [1] Yinya Huang, Meng Fang, Yu Cao, Liwei Wang, and Xiaodan Liang. DAGN: Discourse-aware graph network for logical reasoning. In NAACL, 2021.
>
> **W4:** The role of logical fact regularization.
>
> **A:** c.f. W1. The performance drop of removing this module indicates that the quality of obtained fact unit is of great importance.
>
> **Q1:** Definitions of types for edges.
>
> **A:** self is the edge type to aggregate self-information. Typically, there are default and reverse edges in the Levi graph after transformation [1]. For example, (E1, R1, E2) would become (E1, default, R1), (R1, default, E2), (R1, reverse, E1), and (E2, reverse, R1). However, this may ignore the type of source and target vertices [2]. Therefore, we introduce default-in/out and reverse-in/out to distinguish. We will clarify this in the Appendix.
>
> [1] Jonathan L Gross, Jay Yellen, and Ping Zhang. Handbook of graph theory. Chapman and Hall/CRC, 2013.
>
> [2] Daniel Beck, Gholamreza Haffari, and Trevor Cohn. Graph-to-sequence learning using gated graph neural networks. In ACL, July 2018.
>
> **Q2:** Figure 5's global edge and what this paper claims as global (typical entity or sentence relation) are confusing.
>
> **A:** Thanks for pointing that out. We will distinguish them.
>
> **Q3:** Update for experimental results to match the newly best result in Table 1.
>
> **A:** Because the newly best results are done with DeBERTa to compare with LReasoner. 77.05% and 44.64% are for RoBERTa (there are more baseline models with RoBERTa) for a more intuitive comparison.

---

> > ### Author Response · Authors · 2021-11-29
> > **Looking forward to further discussions!**
> >
> > Dear reviewer UKpM,
> >
> > Thanks for your time and effort in reviewing our work for the second time lol. In the last three weeks, we actively interacted with all reviewers, including detailed clarification for our contribution and methods (Section 3.2 and Appendix A/B), reporting details for the model (Appendix D), and adding further analysis (Appendix C) to address the issues raised in yours and other reviewers’ comments. Most reviewers agreed on the contribution and the significant performance of Focal Reasoner.
> >
> > Would you mind letting us know if our response has addressed your concerns? If you have any additional questions or comments, we would be glad to have further discussions.
> >
> > Thanks,
> >
> > Paper 2553 authors

---

> > ### Comment · Reviewer_UKpM · 2021-11-29
> > **Thank you for your response.**
> >
> > **Positives**:
> > - I agree with the novelty of hierarchical graph structure: organize the fact units by supergraph, which is a hierarchical modeling method. (could authors make sure there wasn't a similar paper before in terms of graph structure for reasoning. I thought I came across something similar, but this could be a misbelief.)
> > - I also think the DeBERTa experiment made the experiment results to be stronger.
> >
> > **Disagreeing in soft manner**:
> > - I do not agree with the concept of "fact unit" being novel, however, I am quite intrigued that dependency parsing can provide such an informative structure.
> > - I believe FocalReasoner should be a model without "logical fact regularization" in order to appreciate the contributions on creating fact-unit and creating a hierarchical graph. Logical fact regularization (LFR) could be an addition to the FocalReasoner (e.g. FocalReasoner + LFR), but I feel having everything together actually makes FocalReaosner's main contribution to be weaker in some sense as everything is lumped together.
> > Could authors consider removing LFT from the main part of the model and presenting it as possible addition that can improve the representation of fact units?  (Reverse of current presentation in some sense)
> > Lastly, could the authors provide the result of DeBERTa without "logical fact regularization" if possible? (I am very sorry for the last-minute request)
> >
> > That being said, I am still at the borderline but I am not particularly against it as well. The paper has strong results and agrees with some of the author's points on novelty. I am changing my score from 5 to 6, but I do have some final doubts on the effect of logical-fact regularization (other concerns are minor) and really wish to see the effect of logical fact regularization on DeBERTa. (Again, sorry for the delayed reply and the last minute request)

---

> > > ### Author Response · Authors · 2021-11-29
> > > **Response to the updates**
> > >
> > > Thanks for your recognition of our response and for raising the score.
> > >
> > > We managed to finish the experiment using DeBERTa model without "logical fact regularization" on ReClor dataset and get the test scores from leaderboard submission. The results are shown in the following.
> > >
> > > Model|Dev|Test|Test-E|Test-H
> > > ---|---|---|---|---
> > > DeBERTa|74.4|68.9|83.4|57.5
> > > Focal Reasoner|78.6|73.3|86.4|63.0
> > > $\quad$ w/o LFR|76.5|72.6|84.6|63.1
> > > LReasoner|74.6|71.8|83.4|62.7
> > >
> > > We can see that Focal Reasoner still obtains superior results by surpassing the baseline model and LReasoner by a large margin without logical fact regularization. We will add this in a future version of this work.
> > >
> > > Thanks for your suggestions again in helping us improve our work!

---

### Official Review · Reviewer_fjem · 2021-11-03

**Correctness:** 3
**Technical Novelty And Significance:** 4
**Empirical Novelty And Significance:** 2
**Recommendation:** 6
**Confidence:** 4

**Main Review:**

## Strengths

* The model seems fairly intuitive. The idea of representing the (entity + relational) structure of a text directly and propagating information through that structure has been prevalent in the IE / RC space for a while but it seems that many have had difficulty getting it to work well. This paper shows improvements which are big enough to indicate that there's probably something to it (though I have a few concerns about this, described in the weaknesses). It seems to me that the key to the success of the model is probably just in the distribution of data that it's applied on. Such approaches may not have worked for earlier datasets (such as SQuAD) simply because they were not suited to teach or test the kind of information aggregating abilities that this model is designed to do.

* The ablations are nice.

## Weaknesses

* I take issue with some of the terminology — use of the terms "fact" and "logical reasoning" are, in my view, a bit misleading as descriptions of what a model like this one is doing. The tuples being extracted from the text are more like "propositions" than "facts" and the operations in the GCN don't seem to have much in common with "logical reasoning" (which normally involves the application of rules from a formal system). I understand there seems to be a bit of precedent for the terms here, as the datasets being used were designed to capture aspects of logical reasoning, but I think it's less correct to reuse them when describing model components such as "Logical Fact Regularization." Something like "propositional structure regularization" might be more appropriate.

* I'm worried that this model's improvements will end up being specific to the distribution in "logic"-focused datasets like ReClor and LogiQA, because of these datasets' focus on aggregating and combining information about a small set of entities in a paragraph. While I understand that this capability is the focus of this work, I think it's important to know if the proposed architecture is too specialized: does it maintain performance over its RoBERTa or DeBERTa baseline if applied to more naturalistic, extractive reading comprehension datasets like Natural Questions, QuoRef, or HotpotQA? If not, is it because of the distribution of text, extractive format, or other factors? It's fine if the the model doesn't end up improving these cases — I think investigating these issues would greatly improve the paper either way.

* I would suggest more caution when it comes to interpretability. Attention weights cannot be relied on to provide explanations of model behavior. To make the interpretability case study in Section 5.3 more believable, I would suggest modifying the attended element and seeing if the answer changes: for example, change "playing football can improve students' academic performance" to "football players are held to higher academic standards than non-athletes" and see if the attention pattern or the answer changes. If it doesn't, then the attention pattern — while still indicative of the information relevant to answer the question - might not be indicative of the kind of logical reasoning we suspect.

* I'm not sure if the findings are statistically robust. All of the test sets here are very small — less than 1000 items. The numbers seem good but I'm not sure what the uncertainty looks like. Please provide confidence intervals for the Focal Reasoner and at least some of the stronger baselines.

* Are the baselines well-tuned? Time and time again, modeling innovations are proposed which ultimately get beaten by a better-tuned baseline down the line. How many experiments did you run for each model variant? Are you reporting the best of several runs? How did the performance of each model vary over runs? Can you say that the distribution of the new model's performance is appreciably better than the baseline models? By how much, and with what confidence?

## Minor comments

* In equations 4 and 5, wouldn't it be better to use $v_\text{predicate}$ instead of $v_\text{relation}$? It's a bit confusing to me to use the word "relation" here since earlier the word is used to refer to the types of edges between vertices in the graph.

**Summary Of The Paper:**

This paper proposes a model architecture for question answering which takes advantage of shallow proposition structure and coreference links using (relational) graph convolutional networks. Subject-predicate-object triples (dubbed "fact units") are extracted from the text using dependency parsing, extra coreference links are added using a coreference system, and the resulting graph is initialized with representations from a contextualizing encoder, passed through a GCN, and recombined with the contextual representations via multi-headed attention. The resulting representations serve as input to the final classification layer for answer prediction, as well as a "logical facts regularization" step which encourages the final representation of the object in each fact unit to align (via cosine) with the sum of the subject and predicate representations.

In experiments on several datasets designed to test logical reasoning, the proposed architecture scores higher than several baselines.

**Summary Of The Review:**

The model seems nice and the results seem positive, but I think the experiments are weak. We don't really learn much from this paper about the strengths and limitations of the proposed model, and it seems plausible that the experimental results are not statistically robust, due to a combination of the small test set size and the limited experimental reporting that is done in the paper.

---

EDIT: Some of my concerns were addressed, particularly the issues regarding reporting more details on the model's performance (number of runs, variance, etc.) which I think were most critical. I am still not satisfied with the discussion of interpretability (which I think should be removed; see discussion below) and I think the paper could have done a better job with experiments demonstrating the relative strengths and weaknesses of the model, but the results that are present seem strong. So I wouldn't say I'm particularly eager to accept the paper but I won't gatekeep it either. Raising my score from a 5 to 6.

---

> ### Author Response · Authors · 2021-11-12
> **Response to Reviewer fjem**
>
> **W1:** Issues with terminology, such as “fact” and “logical reasoning”.
>
> **A:** Thanks for the insightful comments. For “fact”, we are also looking for better terms that make as much sense as possible to express this proposed new concept. We will carefully take your advice into consideration, and make additional notes in the paper to let readers be aware of the term issue when introducing “fact” correspondingly. For “logical reasoning”, as described in Section 1, “we concern a logical reasoning QA task”, we just follow the terminology developed in the MRC community [1,2], but we agree that this is not rigorous even though many MRC literature used the terms in this way as you have observed. We will clarify this correspondingly.
>
>
> **W2:** Model's improvements will end up being specific to the distribution in "logic"-focused datasets
>
> **A:** Thanks for the comments. In fact, we have run experiments on multi-turn dialogue comprehension dataset MuTual which is not a "logic"-focused dataset, to verify the generalization of our model.
>
> **W3:** Caution to interpretability
>
> **A:** Thanks for the valuable suggestion. We have modified the setting in Section 5.3 accordingly and represent the results in Appendix E.
>
> **W4&5:** Robustness for the statistics of Focal Reasoner and baseline models.
>
> **A:** Thanks for your suggestions. All the results for the dev set of Focal Reasoner are the averaged results run with 5 random seeds. We have also provided the averaged results and variance in the Analysis section.
> We took the statistics of baseline models from their official papers [1, 2, 3], and reproduced the results as reported. So, there should be no issue of “not a well-tuned baseline”. Also, there are no official reports regarding averaged results and variance for the baseline models (they do not claim the results are averaged or the best either), and some of the works cannot be reproduced (e.g., LReasoner). We will report the detailed statistics of performance variance in the paper revision. Given the substantial gains over the baselines (+4.3 w/ DeBERTa on the LogiQA test set) and the public models (+2.5 compared with LReasoner) under the same backbone, our model is appreciably better.
>
> [1] Weihao Yu, Zihang Jiang, Yanfei Dong, and Jiashi Feng. Reclor: A reading comprehension dataset requiring logical reasoning. In International Conference on Learning Representations (ICLR), April 2020.
>
> [2] Yinya Huang, Meng Fang, Yu Cao, Liwei Wang, and Xiaodan Liang. DAGN: Discourse-aware graph network for logical reasoning. In NAACL, 2021.
>
> [3] Siyuan Wang, Wanjun Zhong, Duyu Tang, Zhongyu Wei, Zhihao Fan, Daxin Jiang, Ming Zhou, and Nan Duan. Logic-driven context extension and data augmentation for logical reasoning of text. arXiv preprint arXiv:2105.03659, 2021.
>
>
> **Q1:** in equations (4) and (5), use $v_{predicate}$ instead of $v_{relation}$.
>
> **A:** Thanks for the great suggestion. We will revise it.

---

> > ### Comment · Reviewer_fjem · 2021-11-17
> > **Some follow-up questions**
> >
> > Thanks for your quick response. I took a look at the revisions and I have a few more questions.
> >
> > On terminology: I saw the addition "Intuitively, fact unit can be seen as a sort of proposition" and I appreciate it but honestly I don't think this is useful to add.. my point was that the name "fact unit" seems odd but if the name isn't changing then it's only more confusing to throw more terms around. Call it whatever you want I guess.
> >
> > On interpretability: I don't see any changes in Sec. 5.3. Also, rereading it, I find it more confusing. It says "students rank 30%" attends strongly to "playing improve performance" but this pair seems to have the lowest edge weight in the graph (or no edge at all). As for the change that you describe in Appendix E, looks good, except I think that as written, the question's correct answer should be B in the changed example. D would only be correct if the "not" was removed, ie, if it says research *had* proved that playing football contributes to academic performance. Note that this is basically symmetric to the original example, as it's a choice between two explanations (academic requirements for football players, versus playing football contributing to intellectual development) of increased academic performance among footballers that are competing to explain each other away.
> >
> > Also, looking closer at Figure 7, I don't understand what's going on on the right side. That part of the figure and the "interactions" mentioned in the ablations seem to indicate features that compare the answers with each other, but I didn't find any description of this in the modeling section. Could you please describe this in the paper? (Or did I just miss this?) And what is the similarity matrix in Figure 7 supposed to illustrate?

---

> > > ### Author Response · Authors · 2021-11-17
> > > **Response to Reviewer fjem**
> > >
> > > Thanks for your prompt feedback!
> > >
> > > For the "Interpretability": Firstly, we’re sorry for the mistake in the demonstration of Figure 7 when manually depicting the attention weights with grayscale [0, 255]. Thanks for pointing that out. We have fixed it according to our experiment logs. Secondly, we do not quite understand why the answer should be B as the question asks for the option that weakens the conclusion. It seems that our model has made the relatively correct option.
> > >
> > > For the “Interactions”: it is a module considering logical relations among options in the setting of QA, and the attention matrix in Figure 7 is supposed to reflect these relationships. We have made it clearer in Section 3.2.
> > >
> > > Thanks again for your suggestions and reviews.

---

> > > > ### Comment · Reviewer_fjem · 2021-11-17
> > > > **Continuing on the 'interpretability' example...**
> > > >
> > > > The passage notes a correlation between higher academic performance (call it $A$) and playing football ($F$) — call the correlation $A \oplus F$. In the original passage (given in Figure 7), the passage explains this correlation with a cause/effect relationship $F \to A$ (playing football increases academic performance). In this case, answer choice A weakens this conclusion most effectively because it establishes causation in the opposite direction ($A \to F$) in noting that high academic performance is required for playing football. This *explains away* the correlation $A \oplus F$, weakening the evidence it provides for the original conclusion that $F \to A$.
> > > >
> > > > The modified case in the Appendix is symmetric. In the passage, the conclusion was changed to indicate that the academic standards required for football are higher (i.e., $A \to F$). Answer D suggests research has *not* proven that football improves academic performance ($\neg \square (F \to A)$). This suggests an *absence of additional evidence* for the correlation $A \oplus F$ which is known in the passage, meaning an explanation is still required and the original explanation ($A \to F$) is no weaker. However, option B (that playing football can maintain a "strong learning energy") provides evidence for the relationship $F \to A$, which contributes to the correlation $A \oplus F$ and explains away the original conclusion ($A \to F$), weakening it.
> > > >
> > > > (Also, where was this example taken from? Is it from the ReClor development set?)
> > > >
> > > > Anyway, I think this amply illustrates why the interpretability case study needs to be removed altogether because it is not meaningful. The reasoning processes that go into solving these questions are not trivial, and attention weights are not in the least bit interpretable. That the attention weights seem to point an answer choice to a relevant part of the passage does not at all support the idea that the model is following the kind of reasoning process that this data is intended to capture.

---

> > > > > ### Author Response · Authors · 2021-11-29
> > > > > **Thanks for the detailed explanation**
> > > > >
> > > > > We appreciate your valuable feedback.
> > > > >
> > > > > This example is taken from the ReClor development set. We will note the detail in the paper.
> > > > >
> > > > > We are also interested in analyzing the relative strengths and weaknesses of our model, given the significant performance improvement. To this end, we try our best to provide a series of case studies and the "interpretability" section to help readers understand what is going on inside the hard-to-interpret GNN. The interpretation section follows the standard practice in previous studies [1,2] of visualizing attention weights. It is supposed to inspire future works by giving a more explicit demonstration of how our model predicts the answer.
> > > > >
> > > > > We will dig deeper for better ways of interpretation. We will update the title as Section 5.3 as “Case Study” after careful consideration to make it more acceptable to readers. And we are glad to have constructive discussions with you.
> > > > >
> > > > > At last, thanks so much for your recognition of our model’s strong performance and the valuable suggestions to improve our work.
> > > > >
> > > > > [1] Yasunaga, Michihiro, et al. "QA-GNN: Reasoning with Language Models and Knowledge Graphs for Question Answering." NAACL 2021.
> > > > >
> > > > > [2] Lin, Bill Yuchen, et al. "Kagnet: Knowledge-aware graph networks for commonsense reasoning." EMNLP 2019.

---

### Official Review · Reviewer_vNnA · 2021-11-07

**Correctness:** 3
**Technical Novelty And Significance:** 3
**Empirical Novelty And Significance:** 3
**Recommendation:** 6
**Confidence:** 3

**Main Review:**

Strengths:
1. The approach is interesting boasts great results on two benchmarks compared to previous approaches.
2. The authors also perform useful analysis on the model which is interesting and would be useful to the community.
3. The approach is, to some extent, described well.
4. The authors also plan to open source their code which would be beneficial to the community.

Concerns, questions, and suggestions:
1. In the related works section, the authors fail to describe and differentiate themselves from the two baselines that they compare against (DAGN and LReasoner).
2. In the related work section, the authors describe a slew of other methods that the model is not compared against. This makes it hard to place their approach in comparison with prior work thus making it hard to understand the novelty of the approach.
3. Perhaps, the authors could add a section describing related works that are used for the tasks described. (The authors claim that the baselines are described in the appendix but that is not the case)


**Summary Of The Paper:**

The authors propose a framework, called Focal Reasoner, to perform logical reasoning to answer questions. The proposed approach first extracts the facts from raw text. The authors describe a fact unit as a triple of (argument, predicate, argument). These collections of facts can be viewed as a graph with predicates as undirected edges between arguments. After forming a subgraph, reasoning is performed by a graph convolution module to predict the correct answer.
The authors show that their approach outperforms existing approaches on two benchmarked datasets.
The authors also perform a detailed analysis that sheds more light on the internal workings of the model.

**Summary Of The Review:**

The approach is interesting but it is hard to understand what aspect is novel compared to other competitive approaches. Perhaps the authors should describe other methods that have been used to solve this task. This would help make it easier for the reader to understand the novel contribution of the paper and also makes the paper more complete by itself.

---

> ### Author Response · Authors · 2021-11-12
> **Response to Reviewer vNnA**
>
> Q: How to place Focal Reasoner compared with previous work (DAGN, LReasoner)?
>
> A: Thanks for your careful review and insightful suggestions.
>
> Compared with the previous studies, our proposed Focal Reasoner enjoys two major merits as follows:
>
> (1)	Broader Knowledge
>
> Compared with DAGN which uses sententious knowledge such as logical connectives (e.g., “because” “however”), Focal Reasoner leverages a broader type of knowledge characterized by “fact unit”, including global knowledge and local knowledge described in the paper as the basis for reasoning.
>
> (2)	More transferable
>
> Compared with LReasoner which manually designs rules to extract logical patterns and perform logical reasoning in a symbolic way, Focal Reasoner is neural-based and manual-free, which is more generalizable, as we verified the effectiveness in various types of tasks including examination-based comprehension and multi-turn dialogue reasoning.
>
> The major contribution of Focal Reasoner is that 1) we propose to use a unified definition of “fact unit”, which contains a broad type of knowledge that form the necessary basis for logical reasoning, and 2) model the broad facts as a supergraph to capture both global connections between facts and the local concepts or actions inside the fact. Compared with other representations such as NE and SRL, “fact unit” is more effective and easy-to-obtain.
>
> We have added these in Section 3.2 and Appendix B.2 accordingly.

---

> > ### Comment · Reviewer_vNnA · 2021-11-29
> > **Thanks for addressing my concerns**
> >
> > After taking a look at your edits and responses to other reviews, I have a better understanding of your contributions and I will raise my score. Thanks!

---

### Author Response · Authors · 2021-11-12
**Submission Update**

We thank all the reviewers so much for the valuable comments on improving the quality of this work. We have updated the paper according to the feedback and our latest evaluations.

The revision primarily includes:

1. We fixed some typo and presentation problems.

2. We reorganized the methodology section to add more details. (See Section 3.2)

3. We add more details regarding graph construction and the definition of edge types. (See Appendix A)

4. We add descriptions for baseline models to help understand the contribution of our work. (See Appendix B.2)

5. We add a comparison w.r.t. the number of parameters of the model to help address the concern of model complexity. (See Appendix C)

6. We report the average and variances of Focal Reasoner to demonstrate the robustness of statistics. (See Appendix D)

7. We further explore the interpretability of Focal Reasoner. (See Appendix E)

8. We have corrected the mistake of representation in Figure 7.

9. We have made clearer the illustration of **interaction** in Section 3.2.

The revisions are highlighted in the newly updated PDF.

---

### Decision · Program_Chairs · 2022-01-20

**Decision:**

Reject

**Comment:**

Strengths:
* Well-written paper
* Strong empirical results on three benchmarks
* Interesting approach of producing semantically augmented LMs using dependency parses to extract svo triples, and finding coreferences between them across multiple sentences

Weaknesses:
* None of the reviewers seem particularly excited about the paper
* Stronger baseline comparisons would have improved the paper
* Authors re-define a lot of terminology, but the novelty of the method is more from the type of graph used to initialize their method, which seems to be a function of OpenIE triplets